# CRISPR/Cas9-Mediated Knockout of miR-130b Affects Mono- and Polyunsaturated Fatty Acid Content via PPARG-PGC1α Axis in Goat Mammary Epithelial Cells

**DOI:** 10.3390/ijms23073640

**Published:** 2022-03-26

**Authors:** Lian Huang, Jun Luo, Ning Song, Wenchang Gao, Lu Zhu, Weiwei Yao

**Affiliations:** Shaanxi Key Laboratory of Molecular Biology for Agriculture, College of Animal Science and Technology, Northwest A&F University, Yangling, Xianyang 712100, China; kin8248806@163.com (L.H.); songning@nwafu.edu.cn (N.S.); gaowenchang666@163.com (W.G.); zl411143710@163.com (L.Z.); yaoweiwei@nwafu.edu.cn (W.Y.)

**Keywords:** CRISPR/Cas9-mediated knockout, miRNA-130b, fatty acid synthesis, PPARG-PGC1α axis

## Abstract

MicroRNA (miRNA)-130b, as a regulator of lipid metabolism in adipose and mammary gland tissues, is actively involved in lipogenesis, but its endogenous role in fatty acid synthesis remains unclear. Here, we aimed to explore the function and underlying mechanism of miR-130b in fatty acid synthesis using the CRISPR/Cas9 system in primary goat mammary epithelial cells (GMEC). A single clone with deletion of 43 nucleotides showed a significant decrease in miR-130b-5p and miR-130b-3p abundances and an increase of target genes *PGC1α* and *PPARG*. In addition, knockout of miR-130b promoted triacylglycerol (TAG) and cholesterol accumulation, and decreased the proportion of monounsaturated fatty acids (MUFA) C16:1, C18:1 and polyunsaturated fatty acids (PUFA) C18:2, C20:3, C20:4, C20:5, C22:6. Similarly, the abundance of fatty acid synthesis genes *ACACA* and *FASN* and transcription regulators SREBP1c and SREBP2 was elevated. Subsequently, interference with *PPARG* instead of *PGC1α* in knockout cells restored the effect of miR-130b knockout, suggesting that *PPARG* is responsible for miR-130b regulating fatty acid synthesis. Moreover, disrupting *PPARG* inhibits *PGC1α* transcription and translation. These results reveal that miR-130b directly targets the PPARG–PGC1α axis, to inhibit fatty acid synthesis in GMEC. In conclusion, miR-130b could be a potential molecular regulator for improving the beneficial fatty acids content in goat milk.

## 1. Introduction

Goat milk has a variety of fatty acids, especially short- and medium-chain fatty acids and unsaturated fatty acids (UFA), which makes it appealing and contributes to its health benefits [1,2]. The UFA is composed of MUFA and PUFA, which both have beneficial health effects [3,4]. The majority of fatty acids are stored in the triacylglycerol, which is the main source of energy, accounting for about ninety-eight percent of milk lipids [5]. Triacylglycerol, together with cholesterol constitute lipid droplets that store energy and components for membrane synthesis and act as hubs for metabolic processes [6]. Cholesterol is an important molecule for proper cellular and systemic functions [7]. Cholesterol is responsible for growth and development throughout life, and disturbed cholesterol balance underlies various diseases, such as cardiovascular, neurodegenerative, and cancer [8]. The MUFA have anti-inflammatory effects, and a high proportion of MUFA in the diet contributes to the effect of preventing cardiovascular, atherosclerosis, and thrombosis diseases [9]. PUFA is a kind of essential fatty acid for maintaining the health of humans and animals, and it is responsible for effects in preventing inflammatory and cardiovascular diseases [10,11].

MicroRNA regulates milk lipid and fatty acids synthesis through targeting the fatty acid metabolism-related genes and signaling pathways, such as the PPARγ and JAK-STAT5 [12,13]. miR-421-5p regulated ovarian growth and development by targeting the MAPK signaling pathway in rats in a light pollution environment [14]. miR-24 repressed HDL uptake and steroidogenesis by directly targeting *SR-B1* in steroidogenic cells, and overexpression of miR-24 increased gene expression related to cholesterol synthesis [15]. In rats, lnc-HC promoted miR-130b-3p expression by regulating the promoter activity, thereby negatively regulating PPARγ expression and suppressing lipid droplets and TAG formation [16]. A previous study showed that bta-miR-130a/b suppressed cellular TAG and lipid droplet accumulation, by directly targeting *PPARG* and *CYP2U1* in bovine preadipocytes [17]. miR-130b promoted hepatic VLDL assembly and secretion by increasing *MTP* expression and TAG mobilization in immortalized human hepatocytes [18]. In GMEC, miR-130b impaired adipogenesis and targeted both the coding sequence and 3′-untranslated regions (UTR) of *PGC1α* [19]. In ruminants, miR-130b targeted *PPARG*, *CYP2U1,* and *PGC1α,* to inhibit TAG and lipid droplet formation. In non-ruminants, miR-130b suppressed lipid droplets and TAG formation, as well as promoting VLDL synthesis through increasing TAG mobilization. Thus, these studies demonstrate the importance of miR-130b in the regulation of lipogenesis.

Peroxisome proliferator-activated receptor-gamma coactivator 1α (PGC1α) could regulate long-chain fatty acid oxidation and lipid metabolism via controlling the gene expression in the mitochondrial fatty acid oxidation and the tricarboxylic acid cycle pathways [20]. In human skeletal muscle cells, PGC1α could induce peroxisomal activity and mitochondrial fatty acids oxidation by a peroxisomal–mitochondrial functional cooperation [21]. *PGC-1* gene Gly482Ser polymorphism affected low-density lipoprotein cholesterol and total cholesterol concentration with exercise training in older people [22]. A novel sheep *PGC1α* polymorphism with 17 bp insertion in the eleventh intron had an dramatic effect on sheep growth traits [23]. The *PGC1α* gene (c.1892 + 19C > T) polymorphism revealed a significant association with C14:0 and SFA of muscle fat in Fleckvieh bulls [24]. Together, these findings demonstrate the crucial role of *PGC1α* in adipogenesis and fatty acid metabolism and show preliminary applications of functional studies in sheep genetic breeding.

Peroxisome proliferator-activated receptor gamma (PPARγ) may begin the adipogenic process and is important for maintaining the adipocyte phenotype with an activating ligand in mammals [25]. Oxidized low-density lipoprotein promoted total and free cholesterol accumulation via the lncRNA AC096664.3/PPAR-γ/ABCG1 signaling pathway in vascular smooth muscle cells [26]. PPARγ activation stimulated its sumoylation and translocation to the cytoplasm, to induce neutral lipid formation in mouse meibocytes [27]. EPA promoted lipid droplet accumulation through activating PPARγ in the ER in human meibomian gland epithelial cell (hMGEC) [28]. The Pro12Ala variant of the *PPARG* gene showed a significant association with the plasma proportion of eicosapentaenoic acid on fasting free fatty acid concentration in humans [29]. PPARγ activates PGC1α via binding to the PPARγ-responsive element (PPRE) in *PGC1α* promoter in mouse brown adipocytes [30].

The miRNA function was mainly dissected through gain- and loss-of-function methods in the current study, and CRISPR/Cas9 technology has become a more desirable methodology to analyze the effects of an miRNA, due to its high efficiency and specificity, especially the stability of miRNA knockout and the effect of maintaining the subtle miRNA equilibrium in cells [31,32]. Thus, knocking out miRNA using CRISPR/Cas9 can elucidate the function of a given miRNA more accurately, and can effectively clarify the downstream molecular mechanism of the miRNA.

In the present research, we hypothesized that miR-130b could regulate lipid and fatty acid synthesis via targeting the PPARG-PGC1α axis in goat mammary cells. To address this hypothesis, we deleted the coding sequence of miR-130b using the CRISPR/Cas9 system. Through characterizing the phenotypic characteristics of the miR-130b knockout cells, we aimed to elucidate the mechanisms whereby miR-130b regulates lipid and fatty acid synthesis. Altogether, this study provides evidence that editing miR-130b is a promising approach for improving fatty acid contents in goat milk.

## 2. Results

### 2.1. Gene-Editing Single Clone Selection and Analysis

Through predicting the sgRNAs which targeted the genomic sequence of pre-miR-130b, sgRNA1 and sgRNA2, targeting of the miR-130b-5p and miR-130b-3p coding sequence was selected (Figure 1A). Then, through miR-130b knockout cell selection, a single clone with 43 nucleotides deleted between the double-strand break (DSB) sites induced by these two sgRNAs was obtained (Figure 1C). Finally, we evaluated the editing efficiency of the DNA in the single clone and found that the PCR product spanning gene-editing site had a 22.5% T7EN1 cleavage efficiency (Figure 1B).

### 2.2. Analysis of Off-Target Effect Induced by sgRNAs

To detect the OT effect induced by the sgRNAs in the single clone, we predicted the OT sites, and six OT sites of sgRNA1 and five OT sites of sgRNA2 were selected (Appendix A). Agarose gel electrophoresis analysis of PCR product spanning these OT sites showed that these DNA bands were specific and sizes were the same in the miR-130b knockout (KO) and control (WT) group (Appendix A). Then, the OT effect of these sites was evaluated using a T7EN1-cleavage assay, and the result revealed that the OT4 sites of sgRNA2 produced the cleavage bands in the control and miR-130b KO cells (Appendix A). Sequencing the PCR product amplified from the OT4 site revealed that there was a single nucleotide deletion downstream of the OT4 site in control and miR-130b KO cells (Appendix A).

### 2.3. Deletion of miR-130b Sequence Inhibits miR-130b-5p and 3p Expression and Increases PGC1α Abundance

To address the effect of the deletion of the 43 nucleotides on the miR-130b-3p and 5p abundances, we detected their expression and found that both were decreased significantly (*p* < 0.001, Figure 2A). Then, we evaluated the influence of the deletion of these nucleotides on miR-130b biosynthesis; the pri-miR-130b abundance increased, while pre-miR-130b decreased (*p* < 0.001, Figure 2B). Subsequently, we assessed miRNAs expression in the cluster and family with miR-130b. The results, shown in Figure 2C revealed that these nucleotides’ deletion did not affect the expression of miR-130a-3p and miR-301b-3p. (*p* > 0.05). Our results show that the deletion of the 43 nucleotides inducing miR-130b KO inhibits miR-130b-3p and 5p expression through suppressing pre-miRNA processing.

Previous research has shown that miR-130b-3p targeted *PGC1α* in goat mammary epithelial cells [19]. We verified this finding in miR-130b KO cells and found that the mRNA abundance was increased (*p* < 0.001, Figure 2D). Consistently, the protein abundance of PGC1α was also elevated upon knocking out miR-130b (*p* < 0.05, Figure 2E, F).

### 2.4. Knockout of miR-130b Promotes Lipid Droplets, TAG, and Cholesterol Synthesis, and Affects Fatty Acid Composition

Lipid droplets, TAG, and cholesterol contents, and the composition of fatty acids were estimated after deleting miR-130b. As a result, the lipid droplet content in the cytoplasm was increased after miR-130b knockout (*p* < 0.01, Figure 3A,B). Likewise, the relative contents of TAG and cholesterol were both increased (*p* < 0.05, Figure 3C,D). Among the total fatty acids detected in GMEC, the percentage of C14:0, C15:0, and C17:0 increased, and C16:0 and C18:0 decreased after miR-130b knockout (*p* < 0.05). Upon knocking out miR-130b, the percentage of MUFA C16:1, C17:1, and C18:1 decreased (*p* < 0.05). Similarly, the percentage of PUFA C18:2, C20:3, C20:4, C20:5, and C22:6 decreased (*p* < 0.05). Based on the above fatty acid results, the percentage of SFA increased, while MUFA and PUFA decreased, after miR-130b knockout (Table 1). These results demonstrate that miR-130b knockout promotes lipid and SFA synthesis and decreases the MUFA and PUFA contents.

### 2.5. Abundance of Fatty Acid Metabolism-Related Gene Is Changed by Knocking out miR-130b

Next, we detected the mRNA abundance of fatty acid metabolism-related genes, thereby elucidating the roles that key enzymes play in the miR-130b knockout mediating lipid and fatty acid synthesis. The abundance of TAG synthesis genes *GPAM*, *AGPAT6,* and *DGAT1*, and lipid droplets formation gene *ADFP* and *XDH* was increased in miR-130b KO cells (*p* < 0.05). Upon knocking out miR-130b, the mRNA abundance of fatty acid desaturation gene *FADS1* increased and *SCD1* decreased, while the fatty acid elongation genes *ELOVL5* and *ELOVL6* both increased (*p* < 0.05). Similarly, the expression of fatty acid activation gene *ACSL1* increased, and fatty acid transport genes *FABP3* and *CD36* also increased (*p* < 0.01). After miR-130b knockout, the expression of fatty acid synthesis genes *ACC* and *FASN* increased (*p* < 0.001). The abundance of fatty acid oxidation genes *HSL*, *CPT1A*, *ACOX1,* and *PPARA* increased and *ATGL* decreased (*p* < 0.05). Upon knocking out miR-130b, the transcription level of *SREBP1c*, *SREBP2,* and *LXRA* elevated, and *INSIG1* and *INSIG2* reduced (*p* < 0.05, Figure 4).

### 2.6. Inhibition of PGC1α Promotes Lipid Droplets, TAG, and Cholesterol Accumulation, and Affects Fatty Acid Composition

It has been verified that *PGC1α* was targeted by miR-130b in GMEC [19]. To address the function of *PGC1α* in miR-130b knockout-inducing lipid and fatty acid synthesis in GMEC, we transfected siPGC1α in control and miR-130b knockout cells. Transfection of siPGC1α decreased the mRNA and protein abundance of *PGC1α* in miR-130b KO cells (*p* < 0.01, Appendix A–C). As a consequence of *PGC1α* silencing, the lipid droplet content was increased (*p* < 0.01, Appendix A). Meanwhile, the relative contents of TAG and cholesterol were both increased (*p* < 0.01, Appendix A). Fatty acid profiling showed that the percentage of C14:0 and C22:6 decreased and C17:0 and C20:5 increased after knocking down *PGC1α* expression in the control GMEC (*p* < 0.05). Upon interfering with *PGC1α* expression in the miR-130b KO group, the percentage of C16:1 recovered and the percentage of C17:0 further increased (*p* < 0.05). Moreover, the percentage of SFA decreased and PUFA increased, both in the control and miR-130b KO cells, after reducing the expression of *PGC1α* (Appendix A).

### 2.7. PPARG Is a Direct Target of miR-130b in Goat Mammary Epithelial Cells

To identify putative miR-130b targets that could contribute to lipid and fatty acid synthesis, miRNA target-prediction tools, including TargetScan 8.0 (https://www.targetscan.org/vert_80/, data: 23 March 2021) and miRWalk 2.0 (http://mirwalk.umm.uni-heidelberg.de/, data: 23 March 2021), were used. Owing to different scoring algorithms and the functional analysis of putative targets, we selected *PPARG* as a candidate target of miR-130b. As shown in Figure 5A,B, we analyzed the mRNA expression of *PPARG* and found the abundance was decreased after overexpressing miR-130b in control cells and increased in the knockout group. To investigate whether *PPARG* binds with miR-130b in the 3′UTR region, the luciferase reporter vectors psiCheck2 with *PPARG* 3′UTR (WT) and the site-mutated *PPARG* 3′UTR (MUT) were constructed (Figure 5C). Then, a dual-luciferase activity assay demonstrated a significant repression of renilla luciferase activity in the presence of miR-130b, while mutation of *PPARG* 3′UTR resulted in unaffected luciferase activity (Figure 5D). Similarly, the protein abundance of PPARγ was more profoundly reduced after miR-130b overexpression in control cells and elevated in miR-130b KO cells (Figure 5E–H). 

### 2.8. Knockout of miR-130b Promotes Lipid Droplets, TAG, Cholesterol Synthesis and Affects Fatty Acid Composition Mainly via Targeting PPARG

To ascertain whether miR-130b knockout mediated lipid and fatty acid synthesis via targeting *PPARG*, miR-130b KO and control cells were transfected with siPPARG or siNC. After interfering with *PPARG* expression in miR-130b KO cells, the mRNA and protein abundances of *PPARG* were both decreased (*p* < 0.05, Figure 6A–C). The effect of *PPARG* silencing rescued miR-130b knockout, inducing an increase of lipid droplet content (*p* < 0.05, Figure 6D–E). In addition, the promotion of TAG and cholesterol synthesis induced by miR-130b knockout could be partially reversed via reducing the expression of *PPARG* (*p* < 0.05, Figure 6F,G).

Corresponding to lipid droplet and TAG content, the fatty acid profile was analyzed after knocking down *PPARG* expression in miR-130b KO cells. The decreased percentage of C15:0 and C17:0 caused by miR-130b knockout was reversed by suppressing *PPARG* expression (*p* < 0.05). Moreover, miR-130b knockout resulted in a percentage increase of C18:1, and decrease of C18:2, which were both rescued through interfering with *PPARG* expression (*p* < 0.05). Based on the fatty acid proportional change, the decreased percentages of SFA and PUFA and increase of MUFA were all partially alleviated after reducing *PPARG* expression (Table 2). Taken together, these results suggest that downregulation of *PPARG* expression can reverse the promotion effect of lipid and fatty acid synthesis induced by miR-130b knockout in GMEC.

### 2.9. Loss of PPARG Decreases the Expression of PGC1α in miR-130b Knockout Cells

In mice, it has been shown that PPARγ bind to the PPRE element in *PGC1α* promoter, thereby regulating the transcription of *PGC1α* [33]. To explore the regulatory role of *PPARG* in knockout cells, we transfected siNC or siPPARG into miR-130b knockout cells to verify this function. Transfection of siPPARG decreased the mRNA and protein abundance of *PPARG* (*p* < 0.05, Figure 6A–C). As a consequence of *PPARG* expression knockdown, the mRNA abundance of *PGC1α* was decreased (*p* < 0.05, Figure 7A). Correspondingly, the protein abundance of PGC1α was also decreased (*p* < 0.05, Figure 7B,C). These data demonstrate that *PPARG* could inhibit *PGC1α* expression, thereby indicating the importance of the PPARG–PGC1α regulatory axis in miR-130b regulating lipid and fatty acid synthesis.

## 3. Discussion

A deep understanding of miRNA will help to shed light on their regulatory mechanisms in physiological processes, and provide a theoretical foundation for their application in vivo. Dissecting the function of a given miRNA in animals can be achieved by the transfection of miRNA inhibitors [34,35]. However, neither the miRNA inhibitor nor short tandem target mimicry (STTM) approaches can guarantee strong silencing efficacies for all miRNAs [36,37]. Thus, a CRISPR/Cas9 based gene-editing system with more profound and consistent knockout efficiency has been utilized in miRNA studies. Previously, we showed that a CRISPR/Cas9-induced single nucleotide deletion around the Drosha cleavage site decreased mature miRNA abundance through suppressing pri-miRNA processing [38]. Hence, in the present study, we applied a duplex editing strategy to delete the coding regions of pre-miR-130b using two sgRNAs that targeted miR-130b-5p and miR-130b-3p. Due to cas9 protein inducing an off-target effect that limited its application [39], we detected the effect in 11 predicted off-target sites induced by these two sgRNAs, and found that all potential off-target sites were unaffected, which agrees with a previous finding in porcine [40].

The miRNA biogenesis involves nuclear DROSHA processing and cytoplasmic processing DICER [41], and besides the important contribution of DICER and essential effect of DROSHA, the sequence and the stem-loop structure folded by the nascent transcript that affect miRNA biogenesis have been characterized [42]. Our current study in knockout cells showed the increase in pri-miR-130b abundance and pre-miR-130b decrease where the sequence of pre-miR-130b was nearly deleted, which confirmed the importance of the pre-miR-130b sequences. To our knowledge, the miRNA duplex strands were both generated from the biogenesis process of primary miRNA [43]; therefore, we detected the abundance of miR-130b-5p and found it was also downregulated. These results reveal that the nucleotide deletion of partial miR-130b-5p, the entire loop, and nearly all miR-130b-3p inhibits the nuclear processing of pri-miR-130b. Moreover, knockout of miR-130b did not affect the expression of miR-301b-3p, indicating no interdependency exists in the miR-130b~301b cluster, which is consistent with a previous study in miR-106~25 and miR-17~92 clusters [44].

The concentration of intracellular TAG, which is made up of fatty acids and glycerol, was increased after knocking out miR-130b in our research, which agrees with previous studies in porcine and mice [45,46]. In agreement with the function of GPAM, AGPAT6, and DGAT1 enzymes in TAG synthesis [47], the mRNA expression of these three genes was upregulated in miR-130b knockout cells. Cholesterol is another main cellular lipid species that is regarded as a necessary constituent of cell membranes and the regulation of multiple cellular processes [7]. SREBP2 and SREBP1a were involved in cholesterol synthesis by controlling key enzyme expression in this process [48]. Thus, the decrease of mRNA abundance of these two transcription factors after knocking out miR-130b increases cholesterol concentration. Based on the concentration alteration of TAG and cholesterol, which involves neutral lipid formation [6], and the increase in mRNA abundance of *ADFP* and *XDH,* contributing to lipid droplet formation and secretion [49], lipid droplet content in the cytoplasm was elevated in this study.

It is well known that most short- and medium-chain fatty acids and palmitate acids (C16:0) are produced via ACACA and FASN enzymes, inducing fatty acid de novo synthesis [50]. The increased content of C14:0, C15:0, and C17:0 could be caused by *ACACA* and *FASN* mRNA upregulation. The acetyl coenzyme A is the common primer in the de novo synthesis for many kinds of fatty acids; thus, the proportional increase of C14:0, C15:0, and C17:0 could result in the decrease of C16:0. As a consequence of the content decrease of C16:0, the percentage of C18:0 elongated from C16:0 was decreased, although the ELVOL6 enzyme expression level was upregulated. Monounsaturated fatty acids were synthesized by *SCD1* encoding desaturase, which introduces a double bond to C16:0, C17:0, and C18:0 [51,52], the content decrease of these MUFA in knockout cells may result from the decrease of *SCD1* mRNA abundance. The PUFA are endogenously biosynthesized through the desaturation of the UFA and elongation process [53], the proportional decrease of C18:2, C20:3, C20:4, C20:5, and C22:6 could be partly explained by the decrease in C16 and C18 content and the overall effects of the desaturation and elongation processes of fatty acids.

*PGC1α* is targeted by miR-130b in GMEC, and PGC1α regulates lipid metabolism and mitochondrial function in many vital organs [54,55]. With this observation, we detected the effect of *PGC1α* in miR-130b regulating lipid and fatty acid synthesis. In agreement with previous studies in primary human skeletal myocytes, GMEC and transgenic mice [56,57], knocking down *PGC1α* expression in control and miR-130b knockout cells could promote the accumulation of lipid droplet, TAG, and cholesterol. Through analyzing the fatty acid composition in cellular triacylglycerol, we found that the proportion of C16:1, C17:0, and C18:2 was increased after suppressing *PGC1α* expression; and combined with similar results in mice [58], these findings show the importance of PGC1α in fatty acid and lipid synthesis.

The *PPARG* plays an integrated role in regulating gene expression in the storage and mobilization of fatty acid; thus, promoting fatty acid storage and lipogenesis in adipose and mammary tissues [59,60]. Previous research showed miR-130b regulated adipose tissue inflammation and insulin tolerance via targeting *PPARG* in mice [61]. In line with this result, our observation revealed that miR-130b reduces the expression of *PPARG* by binding to its 3′UTR region. Subsequently, we tried to study the role of *PPARG* in miR-130b regulating lipid and fatty acids synthesis. Following studies in humans and mice [62,63], we showed an opposite phenotype for knockout of miR-130b and silencing of *PPARG* in GMEC. More importantly, the effects of miR-130b knockout on lipid droplet, TAG, and cholesterol accumulation were rescued by silencing of *PPARG*. Moreover, the proportional decrease of C15:0, C17:0, and C18:2, and C18:1 increase was partly reversed by interfering with *PPARG* expression. As *PPARG* could promote *PGC1α* expression through binding to its promoter region [30], we estimated the effect of *PPARG* silencing on the mRNA and protein abundances of *PGC1α*, and detected a significant abundance decrease of *PGC1α*. These results indicate that PPARγ plays a crucial role in the effects of miR-130b in GMEC, and they revealed the existence of the PPARG–PGC1α axis in miR-130b regulating lipid and fatty acid synthesis.

## 4. Materials and Methods

### 4.1. Ethics Statement

The experimental procedure for dairy goats was approved by the Institutional Animal Care and Use Committee of the Northwest A&F University, Yangling, Shaanxi, China (permit number: 15–516, date: 13 September 2015).

### 4.2. Mammary Collection and Cell

The mammary gland tissue was separated from mammary gland biopsies of 3 healthy three-year-old Xinong Saanen dairy goats during peak lactation (120 days postpartum). Mammary tissues (1–2 g) were washed with diethylpyrocarbonate (DEPC)-treated phosphate-buffered saline (PBS) and trimmed of fat tissue. The goat mammary epithelial cells were separated from the mammary alveoli and purified and cultured individually for 5 passages, to obtain the pure epithelial cells for subsequent experiments. The detailed procedures of isolation and purification of primary GMEC were published previously [64]. Cells from each goat were incubated in an environment of 5% CO_2_ at 37 °C with a basal medium. The basal medium contained 90% DMEM/F12 (SH30023−01, Hyclone, Logan, UT, USA), 5 μg/mL bovine insulin (16634, Sigma, St. Louis, MO, USA), 100 U/mL penicillin/streptomycin, 10 ng/mL epidermal growth factor (PHG0311, Invitrogen, Waltham, MA, USA), 5 μg/mL hydrocortisone (H0888, Sigma), and 10% fetal bovine serum (10099-141, Invitrogen). To induce lactation, 2 μg/mL prolactin (L6520, Sigma, St. Louis, MO, USA) was used for two days prior to subsequent experiments.

### 4.3. Construction of Cas9-sgRNAs and PPARG 3′UTR Vectors

The sgRNAs targeting the pre-miRNA genomic sequence of miR-130b were predicted with the Guide RNA design website (http://chopchop.cbu.uib.no/, data: 9 August 2020). The Cas9-sgRNA1-sgRNA2 vector was constructed with two sgRNAs that targeted both ends of the pre-miR-130b genomic sequence. sgRNA1 and sgRNA2 were synthesized and annealed to construct double-strand oligonucleotides. Then, the double-strand sgRNA1 and sgRNA2 were inserted into the pSpCas9 (BB)-2A-Puro plasmid (62988, Addgene, Cambridge, UK) at the *Bbs*I site, to construct the PX459-sgRNA1 and PX459-sgRNA2 vectors, respectively. Next, The U6-sgRNA2-tracRNA fragment in the PX459-sgRNA2 vector was amplified by PCR reaction using the primers: F: 5′-CACCTCTAGAGAGGGCCTATTTCCCATGATTCCTTCATAT-3′, R: 5′-CACCGGTACCAAAAAAGC-ACCGACTCGGTGCCACTTTTTC-3′, and, subsequently, cloned in the PX459-sgRNA1 vector at *Xba*I-*Kpn*I sites to construct the Cas9-sgRNA1-sgRNA2 vector. 

The 3′UTR region of peroxisome proliferator-activated receptor-gamma (PPARγ) coding sequence (*PPARG*) containing miR-130b binding sites was inserted into a psiCHECK2 vector to produce the psi-WT-PPARG vector using the primers F: 5′-CCGCTCGAGCAGAGAAGTCCGAGCTCA-3′, R: 5′-AAGGAAAAAAGCGGCCGCCC-TCAAAATAATAGTGCAAC-3′. The psi-MUT-PPARG plasmid containing site-directed mutagenesis of binding sequences of miR-130b seed region in PPARG 3′UTR was constructed using the primers F: 5′-CCGCTCGAGCAGAGAAGTCCGAGCTCA-3′, R: ATAAGAATGCGGCCGCCCTCAAAATAATGTGTTGGCTGGAGAAGGAAGATGTC-GC-3′ through overlapping PCR. 

### 4.4. Plasmid or RNA Transfection, Single Clone Selection, and T7EN1 Cleavage Assay

The GMEC was transfected with PX459 or Cas9-sgRNA1-sgRNA2 vector at the confluence of 70 to 80% with Lipofectamine 2000 reagent (11668019, Invitrogen), following the manufacturer’s recommendation. After transient transfecting for forty-eight hours, these cells were subjected to puromycin (puro) (1.0 μg/mL) selection, following single-cell clone selection. The DNA fragment spanning pre-miR-130b genomic sequence was amplified by PCR reaction with the primers: F 5΄-ACTTGTGCTTGCTTTCCG-3′, R 5′-GAGCCCCTCGTCCTACCTG-3′. The fragment was subjected to T7EN1-cleavage assay and sanger sequencing for gene editing detection. Detailed procedures of single-cell clone selection and gene editing detection were mentioned in our previous publications [38,65].

The 3′UTR region of *PPARG* was predicted to bind with miR-130b from the TargetScan database. To overexpress miR-130b, GMEC was transfected with miR-130b mimic (50 nM, RiboBio, Guangzhou, China) or its scramble (mimic-NC) or at 70–80% confluence using LipofectamineTM RNAiMAX (13778150, Invitrogen), following the manufacturer’s instructions.

To reduce *PPARG* expression, small interfering RNA (siRNA, siPPARG) were synthesized from the Gene Pharma corporation (Shanghai, China). The GMEC was transfected with three siRNAs (siPPARG-195, siPPARG-980, and siPPARG-1164) to detect the knockdown efficiency (Appendix A). Cells were transfected with siPPARG, siPGC1α or siNC (50 nM) using LipofectamineTM RNAiMAX. The sequence of siPGC1α has previously been reported [56], and the sequences of all siRNAs are displayed in Appendix A.

### 4.5. Analysis of Off-Target (OT) Effect in Single Clone

The OT effect of Cas9 protein induced by sgRNAs was predicted with CRISPR RGEN Tools (http://www.rgenome.net/cas-offinder/, data: 27 October 2020). Eleven OT sites were selected and the DNA fragment spanning the OT site was amplified using the genome extracted from the single clone. Then, the DNA fragment was used for OT effect analysis by T7EN1 assay and the subsequent sequencing. The primer used for OT effect analysis is displayed in Appendix A.

### 4.6. Reverse Transcription Quantitative PCR (RT-qPCR)

The GMEC was seeded in the 12-well plates and transfected with reagent at 70–80% confluence. After forty-eight-h transfection, cells were collected and total RNA was isolated with TRIzol reagent. The quality of RNA was evaluated through detecting the concentration, and the ratio of OD260/OD280 was 1.8–2.0 and OD260/OD230 was 2.0–2.2. The RNA integrity was determined through analysis of 18S and 28S rRNA by agarose gel electrophoresis, and the 18S/28S ratio was 1:2.

Relative miRNA expression was examined using the S-Poly(T) Plus method from 0.5 μg RNA, following the manufacturer’s protocol [66]. 18S rRNA was detected as endogenous control, and the abundance of miRNA was determined using the 2^−ΔΔCt^ method. The primer sequence is displayed in Appendix A.

mRNA expression was evaluated with the PrimeScriptTM RT Kit, following TB GreenTM *Premix Ex Taq*^TM^ II (RR820A, Perfect Real Time, Takara Bio Inc, Otsu, Japan), according to the manufacturer’s instructions. Genome in RNA was cleaned with the gDNA eraser in the PrimeScript^TM^ RT reagent Kit. RNA was reverse transcripted and then tested to detect the mRNA expression by qPCR. The mRNA abundance was detected via the 2^−ΔΔCt^ method normalized to ubiquitously expressed transcript (*UXT*) and ribosomal protein S9 (*RPS9*). All primers in the mRNA detection are listed in Appendix A.

### 4.7. Western Blot

The GMEC was plated and transfected for 48 h. Then, cells were lysed and total proteins were collected using RIPA reagent (R0010, Solarbio, Beijing, China) containing protease inhibitor cocktail. The concentration of proteins was examined with the BCA assay reagent. Detailed Western blot procedures were mentioned previously [38]. The primary antibodies included goat anti-PGC1α (ab106814, Abcam, Cambridge, UK; 1:500), rabbit anti-PPARγ (16643-1-AP, Proteintech Group, Wuhan, China; 1:2000), and mouse anti-*β*-Actin (CW0096, CW Biotech, Beijing, China; 1:2000). The secondary antibodies contained goat anti-mouse and anti-rabbit IgG (CW0102, 0103, CW Biotech, Beijing, China; 1:4000) and rabbit anti-goat-IgG (D110117, Sangon Biotech, Shanghai, China; 1:4000). After washing three times with TBST, the signals of protein were examined with a chemiluminescent (ECL) system. The content of protein was measured using Image J and normalized by comparison to *β*-Actin.

### 4.8. BODIPY™ 493/503 and DAPI Staining

The GMEC was seeded in the 12-well plates. Then, cells were harvested for fixing by 4% paraformaldehyde at 4 °C for 30 min. Lipid droplets in the cytoplasm were examined by 0.1% BODIPY 493/503 (Invitrogen) staining for 30 min. Nucleus was counterstained with DAPI solution (C1006, Beyotime, Jiangsu, China) for 10 min. Each step of staining was followed by washing three times with PBS. The image of lipid droplets and nucleus was acquired by a Cell imaging detector (BioTek Instruments Inc, Winooski, VT, USA). The relative value of BODIPY was quantified through normalizing to DAPI.

### 4.9. Total TAG and Cholesterol Assays

Cells were harvested for TAG and cholesterol content determination. The intracellular TAG and cholesterol concentrations were quantified using a triacylglycerol and cholesterol kit (E1013, E1015, Applypen Technologies Inc, Beijing, China), following the manufacturer’s recommendations. Protein concentration was examined with a BCA assay kit. Relative TAG and cholesterol concentration was determined by comparing them to the protein concentration and expressed as μg/mg protein. Detailed procedures have previously been reported [50]. 

### 4.10. Fatty Acid Composition Analysis

The GMEC was plated in a 60 mm dish, and cells were harvested for fatty acid extraction, after incubating for forty-eight hours. Cellular fatty acid was extracted and methyl esterified with sulfuric acid/methanol (2.5%). The procedures were conducted as published previously [67]. Fatty acid extraction and analysis processes were performed with three samples from three goats and repeated three times. The proportions of each fatty acid were calculated as the ratio of total fatty acid. Prior to the fatty acid analysis, we analyzed the Supelco 37 Component FAME Mix (CRM47885, Sigma) for peak analysis.

### 4.11. Luciferase Reporter Assay

For target gene identification, GMEC was seeded at the confluence of 70–80%. Then, GMEC was transfected with a psi-WT-PPARG or psi-MUT-PPARG vector. Six hours after transfection, miR-130b mimic or scramble was employed to transfect these cells for forty-eight hours. After that, cells were lysed to determine the Renilla and firefly luciferase activity with the application of a Luciferase assay kit. The renilla luciferase activity of *PPARG* 3′UTR was determined by comparing with firefly luciferase activity.

### 4.12. Statistical Analysis

Data are expressed as mean ± SEM. The experiments are conducted in three biological replicates from three goats and repeated three times. Statistical significance was calculated by applying a two-tailed Student’s t-test or one-way ANOVA in multiple comparisons with SPSS 19.0. Differences were declared statistical significance at *p* < 0.05.

## 5. Conclusions

In conclusion, miR-130b could inhibit lipid droplet, TAG, and cholesterol accumulation and affected fatty acid composition, especially MUFA and PUFA proportion, in goat mammary epithelial cells. Functional analyses indicated that the PPARG–PGC1α axis, made up of two direct targets of miR-130b, might contribute to miR-130b-mediating lipid and fatty acid synthesis. These findings suggest the important role of the miR-130b-PPARG–PGC1α axis in those processes in GMEC. Further studies are necessary to elaborate the molecular mechanism of *PPARG* and *PGC1α,* along with their downstream targets, to provide a theoretical frame for improving the composition of beneficial fatty acids in goat milk.

## Figures and Tables

**Figure 1 ijms-23-03640-f001:**
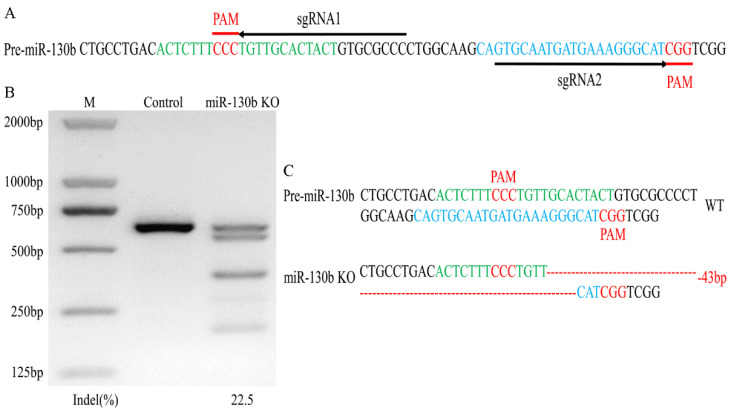
sgRNA targets the pre-miR-130b coding sequence in goat, and gene-editing analysis of single-cell clone. (**A**) Pre-miR-130b genomic sequence and its corresponding sgRNAs selection. sgRNA1 and sgRNA2 are displayed as horizontal arrowheads (black), and PAM sequences are underlined and in red. miR-130b-5p and miR-130b-3p genomic sequences are shown in green and blue, respectively. (**B**) Cleavage efficiency of the PCR product spanning pre-miR-130b sequence was detected via T7EN1 assay and the intensity of bands was calculated by ImageJ. (**C**) DNA sequencing confirmed the indels sequence in miR-130b alleles in the single clone. Wild type (WT) and nucleotide deletion (-) are displayed on the right and deleted nucleotides are indicated with a red dotted line.

**Figure 2 ijms-23-03640-f002:**
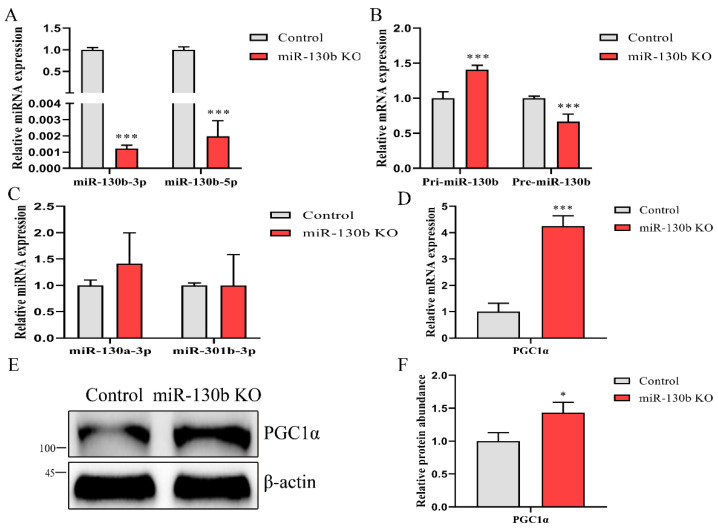
Knocking out miR-130b decreases miR-130b-3p and 5p abundance and promotes *PGC1α* expression. (**A**) Detecting the abundance of miR-130b-3p and 5p by RT-qPCR in control and miR-130b KO cells. (**B**) Abundance analysis of pri-miR-130b and pre-miR-130b in these two cells. (**C**) Expression detection of miR-130a-3p and miR-301b-3p in KO cells. (**D**) mRNA expression analysis of *PGC1α* in miR-130b KO and control cells. (**E**,**F**) Detection of PGC1α protein abundance in control and KO cells by Western blot. Data are shown as mean ± SEM. *p* < 0.05 and *p* < 0.001 are indicated as * and ***.

**Figure 3 ijms-23-03640-f003:**
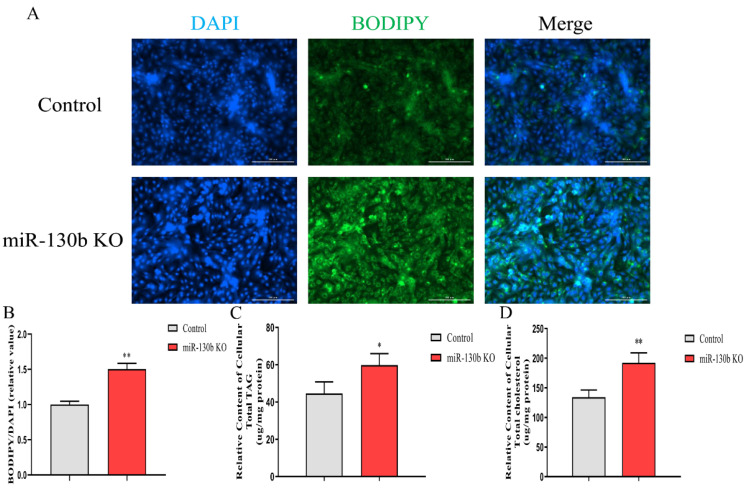
Knockout of miR-130b promotes cellular lipid droplets, TAG and cholesterol accumulation. Detection of lipid droplets (**A**,**B**), TAG (**C**), and cholesterol (**D**) contents in miR-130b KO and control cells. Data are presented as mean ± SEM. *p* < 0.05 and *p* < 0.01 are indicated as * and **.

**Figure 4 ijms-23-03640-f004:**
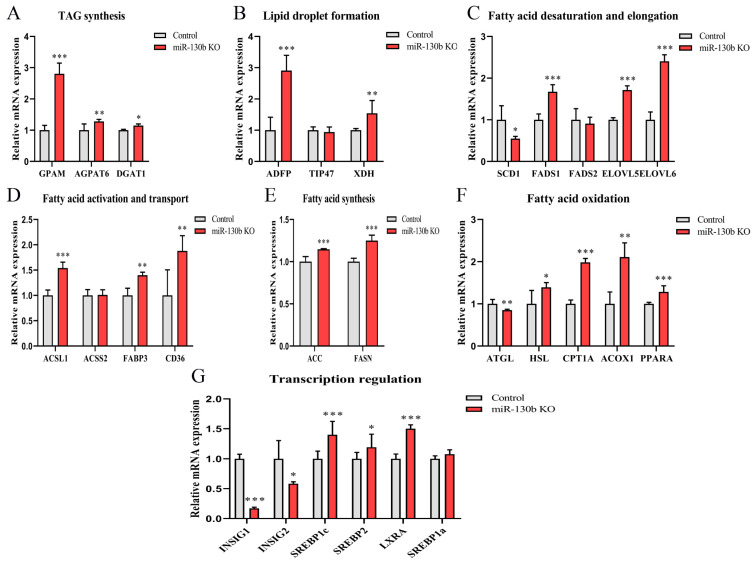
miR-130b knockout promotes the gene expression associated with fatty acid metabolism. mRNA abundance detection of the gene of TAG synthesis (**A**), lipid droplet formation (**B**), fatty acid desaturation and elongation (**C**), fatty acid activation and transport (**D**), fatty acid synthesis (**E**), fatty acid oxidation (**F**), and transcription regulation (**G**) in miR-130b KO and control cells. Data are shown as mean ± SEM. *p* < 0.05, *p* < 0.01, and *p* < 0.001 are indicated as *, ** and ***, respectively.

**Figure 5 ijms-23-03640-f005:**
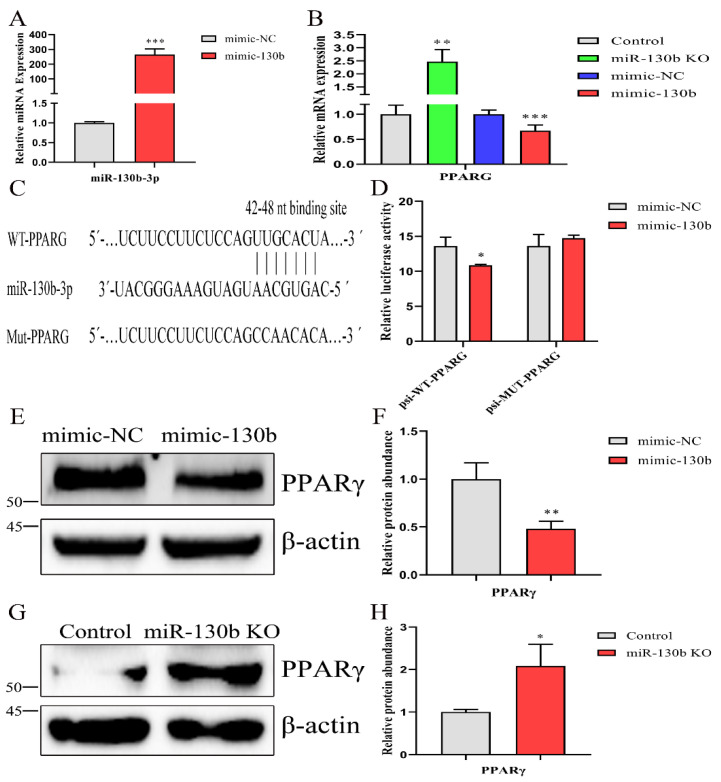
miR-130b-3p specifically targets *PPARG* 3′UTR. (**A**) Detection of the abundance of miR-130b-3p by RT-qPCR after mimic-130b transfection in control cells. (**B**) mRNA expression analysis of *PPARG* after overexpressing miR-130b in control cells or knocking out miR-130b. (**C**) The binding site of miR-130b-3p seed sequence in 3′UTR region of *PPARG* and site-directed mutation of the binding site in *PPARG* 3′UTR. WT-PPARG represents the wild-type (WT) *PPARG* 3′UTR (42–48) sequence; MUT-PPARG represents the mutational *PPARG* 3′UTR sequence. (**D**) Detection of the luciferase activity in GMEC co-transfected with the miRNA (mimic-130b or mimic-NV) and dual-luciferase reporter plasmid (psi-WT-PPARG or psi-MUT-PPARG). (**E**,**F**) Detection of PPARγ protein abundance by Western blot after mimic-130b transfection in control cells. (**G**,**H**) Protein abundance detection of PPARγ in miR-130b KO and control cells. Data are displayed as mean ± SEM. *p* < 0.05, *p* < 0.01 and *p* < 0.001 are indicated as *, ** and ***, respectively.

**Figure 6 ijms-23-03640-f006:**
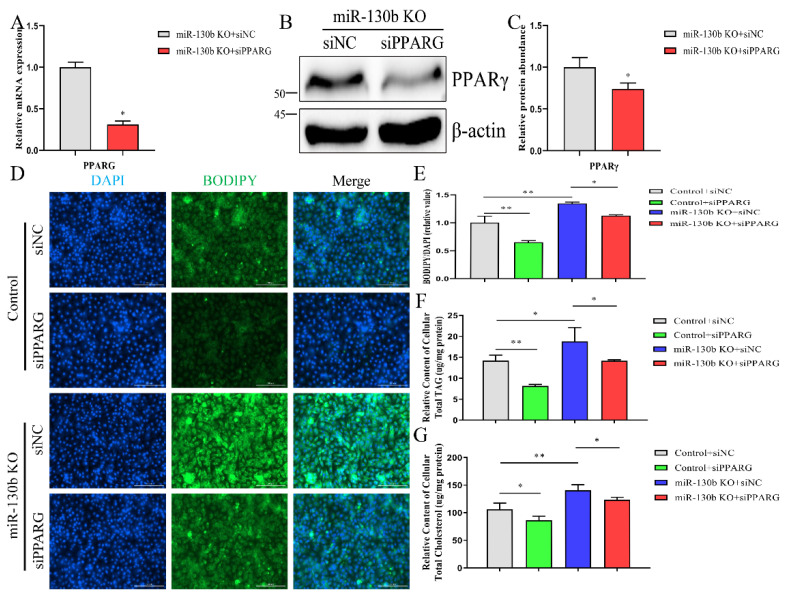
Knocking out miR-130b promotes lipid droplets, TAG, and cholesterol synthesis via targeting *PPARG*. (**A**) Abundance analysis of *PPARG* mRNA after interfering with its expression in miR-130b KO cells. (**B**,**C**) Detection of PPARγ protein abundance in miR-130b KO cells transfected with siPPARG or siNC. (**D**–**G**) Detection of the contents of lipid droplets (**D**,**E**), TAG (**F**), and cholesterol (**G**) in miR-130b KO and control cells transfected with *PPARG* siRNA or NC. Data are displayed as mean ± SEM. *p* < 0.05 and *p* < 0.01 are indicated as * and **.

**Figure 7 ijms-23-03640-f007:**
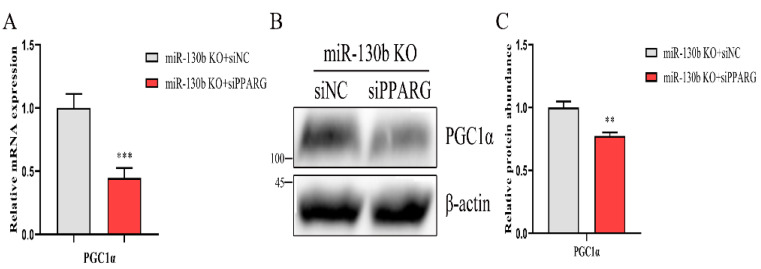
Silencing the expression of *PPARG* inhibits the *PGC1α* expression in knockout cells. (**A**) Abundance detection of *PGC1α* mRNA by RT-qPCR after reducing *PPARG* expression. (**B**,**C**) Detection of the protein abundance of PGC1α in GMEC transfected with *PPARG* siRNA or NC. Data are displayed as mean ± SEM. *p* < 0.01 and *p* < 0.001 are indicated as ** and ***.

**Table 1 ijms-23-03640-t001:** Effects of miR-130b Knockout on Fatty Acid Composition in GMEC ^1^.

Fatty Acid (%)	Control	miR-130b KO
C14:0	4.01 ± 0.00 ^B^	6.36 ± 0.56 ^A^
C15:0	0.67 ± 0.03 ^b^	0.73 ± 0.00 ^a^
C16:0	46.54 ± 0.23 ^a^	46.03 ± 0.12 ^b^
C16:1	1.69 ± 0.04 ^A^	1.37 ± 0.00 ^B^
C17:0	1.10 ± 0.05 ^B^	1.15 ± 0.01 ^A^
C17:1	0.44 ± 0.06 ^a^	0.41 ± 0.02 ^b^
C18:0	27.84 ± 0.10 ^a^	27.14 ± 0.26 ^b^
C18:1	12.93 ± 0.08 ^A^	12.64 ± 0.05 ^B^
C18:2	1.35 ± 0.00 ^A^	1.18 ± 0.04 ^B^
C20:0	0.32 ± 0.04	0.29 ± 0.01
C20:3	0.52 ± 0.01 ^b^	0.49 ± 0.04 ^a^
C20:4	1.60 ± 0.11 ^a^	1.38 ± 0.08 ^b^
C20:5	0.53 ± 0.05 ^B^	0.45 ± 0.02 ^A^
C22:6	0.46 ± 0.02 ^A^	0.38 ± 0.01 ^B^
SFA	80.48	81.70
MUFA	15.06	14.41
PUFA	4.46	3.89

^1^ Fatty acid data are expressed as the ratio of total fatty acids detected in cells. Statistical significance of the difference between miR-130b knockout (KO) and corresponding control GMEC is as follows: *p* < 0.05 and *p* < 0.01 are shown as lower case letters and upper case letters, respectively. Data are shown as mean ± SEM.

**Table 2 ijms-23-03640-t002:** Effects of *PPARG* Silencing on miR-130b Knockout Mediating Fatty Acid Composition Change.

Fatty Acid (%)	Control + siNC	Control + siPPARG	miR-130b KO + siNC	miR-130b KO + siPPARG
C14:0	3.58 ± 0.32 ^a^	3.08 ± 0.09 ^b^	3.76 ± 0.13 ^a^	3.71 ± 0.18 ^a^
C15:0	0.43 ± 0.01 ^a^	0.45 ± 0.01 ^a^	0.36 ± 0.01 ^c^	0.39 ± 0.01 ^b^
C16:0	51.29 ± 0.60 ^b^	52.47 ± 0.14 ^a^	50.48 ± 0.46 ^b^	50.37 ± 0.55 ^b^
C16:1	2.11 ± 0.07 ^a^	2.05 ± 0.07 ^a^	1.88 ± 0.03 ^b^	1.86 ± 0.09 ^b^
C17:0	0.72 ± 0.02 ^b^	0.79 ± 0.01 ^a^	0.61 ± 0.03 ^c^	0.71 ± 0.02b ^b^
C17:1	0.43 ± 0.01 ^ab^	0.43 ± 0.02 ^a^	0.37 ± 0.01 ^c^	0.39 ± 0.03 ^bc^
C18:0	21.95 ± 0.49 ^b^	22.98 ± 0.27 ^a^	22.19 ± 0.38 ^ab^	22.98 ± 0.73 ^a^
C18:1	15.38 ± 0.19 ^c^	14.09 ± 0.27 ^d^	16.71 ± 0.14 ^a^	15.78 ± 0.13 ^b^
C18:2	0.98 ± 0.03 ^a^	0.91 ± 0.03 ^b^	0.83 ± 0.01 ^c^	0.88 ± 0.01 ^b^
C20:0	0.31 ± 0.02 ^b^	0.25 ± 0.02 ^c^	0.37 ± 0.02 ^a^	0.38 ± 0.03 ^a^
C20:3	0.37 ± 0.03 ^ab^	0.35 ± 0.01 ^b^	0.39 ± 0.02 ^a^	0.41 ± 0.02 ^a^
C20:4	1.59 ± 0.08 ^a^	1.42 ± 0.05 ^b^	1.37 ± 0.03 ^b^	1.43 ± 0.07 ^b^
C20:5	0.54 ± 0.01 ^a^	0.51 ± 0.02 ^a^	0.34 ± 0.02 ^b^	0.36 ± 0.01 ^b^
C22:6	0.32 ± 0.00 ^a^	0.22 ± 0.01 ^b^	0.33 ± 0.02 ^a^	0.34 ± 0.01 ^a^
SFA	78.28	80.01	77.28	78.54
MUFA	17.91	16.58	18.97	18.03
PUFA	3.81	3.41	3.25	3.43

Statistical significance is as follows: *p* < 0.05 is shown as lower case letters. Data are expressed as mean ± SEM.

## Data Availability

Not applicable.

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
