# Peer review of "CRISPR/Cas9-Mediated Knockout of miR-130b Affects Mono- and Polyunsaturated Fatty Acid Content via PPARG-PGC1α Axis in Goat Mammary Epithelial Cells"

_ijms, 2022, doi:10.3390/ijms23073640_

Round 1

Reviewer 1 Report

In the manuscript by Huang et al., the authors study the role of miR-130b in fatty acid synthesis in goat mammary epithelial cells. The authors utilize CRISPR/Cas9 system to KO miR-130b in vitro and uncover the underlying mechanism that involves PPARG-PGC1α as a major axis regulating this effect. Authors have exploited various silencing and overexpression approaches to confirm their findings. However, there are several concerns that are needed to be addressed before the manuscript can be taken into consideration.

Major concerns

  • The title of manuscript is misleading “CRISPR/Cas9-mediated Knockout of miR-130b Reduces Mono-and Polyunsaturated Fatty Acid…”. The authors claim that both MUFA and PUFA were decreased in miR-130b KO cells as compared to control. However, the claim is not fully supported by the data presented. Firstly, the decrease is very marginal in miR-130b KO cells (~5% reduction in MUFA and ~13% in PUFA) as shown in Table 1. Secondly it is intriguing and noteworthy that in Table 2 instead of decrease in MUFA in Control+siNC vs miR-130b KO+siNC the MUFA values are increased. It is strange that scramble siRNA could reverse the fatty acid content in these cells. Similarly, in Supplementary Table S5, the Control+siNC vs miR-130b KO+siNC show that PUFA are increased instead of decreasing in miR-130b KO cells as compared to control cells. In fact, in Table 3 and Supplementary Table S5, it is confusing the just scramble siRNA could show opposite % of C14:0, C15:0, C18:0, C20:3, C20:0, C22:6 in miR-130b KO cells as compared to control when compared to these values in control vs miR-130b KO cells in Table 1. Authors should clarify this.
  • For fatty acid composition analysis since most of the differences in the % of fatty acids are not as dramatic, it would be essential to mention number of samples and times this analysis was performed.
  • In supplementary figure S3, it appears that siPGC1a in control cells itself is sufficient to induce lipid droplets (supplementary figure S3E) and total cellular cholesterol levels (supplementary figure S3G) to comparable levels of miR-130b KO and no further additive effect is observed in miR-130b KO cells transfected with siPGC1α. In other words, the difference observed in Control+siNC vs miR-130b KO+siNC are not observed in Control+siPGC1a vs miR-130b KO+siPGC1a Authors should comment on this.
  • The description of figure 5 in result section (page 7 line 185-197) is very disorganized and difficult to follow. Authors should re-write and re-arrange the data in sequence.
  • Also, in figure 5 D-F, it is not clear that the overexpression of mimic-NC and mimic-130b is performed in control cells or miR-130b KO cells? If the overexpression is done in miR-130b KO cells, it would be critical to show effect of overexpressing mimic-130b in control cells.
  • Again, in figure 7, it is not mentioned (in results or in figure legends) whether the silencing of PPARG was performed in miR-130b KO cells to analyze the downstream PGC1α expression. To dissect importance of PPARG-PGC1α regulatory axis in miR-130b, these experiments need to be done in miR-130b KO cells.

Minor concerns

  • Line 31 and 32 in introduction are exactly same.
  • In materials and methods, line 380-384 seems not relevant.
  • Authors should carefully revise the entire manuscript for grammatical and typographical errors.

Reviewer 2 Report

The authors needs to revise the introduction to discuss more about the background, and update the literature references

Please add at least 2 master tables in order to compare the results of the literature

Improve the quality of the figures

Round 2

Reviewer 1 Report

Authors have nicely addressed all my concerns/comments. I don’t have further concerns.